# Interpretable Steering of Large Language Models with Feature Guided Activation Additions

**Samuel Soo**[1], **Chen Guang**[2], **Chandrasekaran Balaganesh**[1], **Wesley Teng**[1], **Tan Guoxian**[1], **Yan Ming**[3]

[1]Raffles Science Institute, Raffles Institution
[2]Nous Research
[3]Centre for Frontier AI Research (CFAR), Agency for Science Technology and Research (A*STAR)
{samuel.soo.ey@gmail.com, guoxian.tan@ri.edu.sg, mingy@cfar.a-star.edu.sg}

## Abstract

Effective and reliable control over Large Language Model behavior is a significant challenge. While activation steering methods, which add steering vectors to a model's hidden states, are a promising approach, existing techniques often lack precision and interpretability in how they influence model outputs. We introduce Feature Guided Activation Additions (FGAA), a novel activation steering method that leverages insights from Contrastive Activation Addition (CAA) and Sparse Autoencoder-Targeted Steering (SAE-TS). By operating in the latent space of a Sparse Autoencoder (SAE) and employing optimization techniques to select desired SAE features, FGAA constructs precise, human-interpretable steering vectors that provide better steering effects while maintaining coherence of steered model outputs. In this regard, evaluations on Gemma-2-2B and Gemma-2-9B models across various steering tasks demonstrate that FGAA outperforms existing steering methods of CAA, SAE decoder steering, and SAE-TS. Our results also highlight important trade-offs between steering scale and general model capabilities that are consistent across all tested steering methods.

## 1 Introduction

The reliable and effective control of Large Language Models (LLMs) has emerged as an increasingly significant challenge in recent years. While researchers have developed various approaches to influence LLM behavior, the limitations of existing methods warrant careful consideration. Fine-tuning (Ouyang et al., 2022) offers some behavioral control but demands substantial computational resources and carefully curated datasets, making it impractical for many applications. Similarly, instruction-based approaches through prompting (Wallace et al., 2024) provide a degree of influence over model outputs but often lack robustness when faced with adversarial inputs or complex tasks. Activation steering has recently gained attention as an alternative methodology that potentially addresses these shortcomings by directly manipulating the model's hidden state representations during inference. This technique involves introducing steering vectors at specific points in the forward pass to guide the model's behavior in desired directions. Nevertheless, current implementations of activation steering face challenges related to interpretability, precision, and consistency which frequently resulting in unpredictable behavioral shifts and degraded output quality that limit their practical utility.

Recent work on SAE-Targeted Steering (SAE-TS) (Chalnev et al., 2024) demonstrated the value of using Sparse Autoencoders (SAEs) to extract targetable features during steering. Building on this and Contrastive Activation Addition (CAA) (Rimsky et al., 2024), we present Feature Guided Activation Additions (FGAA).

We evaluate FGAA against multiple baselines, including traditional activation steering, SAE decoder steering, and SAE-TS, across various steering tasks on both Gemma-2-2B and Gemma-2-9B models (Rivière et al., 2024). Our experiments demonstrate that FGAA achieves superior performance in both

steering effectiveness and output coherence, particularly in complex steering tasks where maintaining text coherence has traditionally been challenging.

This work contributes to the field of controlled text generation in several ways:

1. We develop a novel method FGAA for constructing steering vectors, harnessing benefits from SAE insights, as well as CAA and SAE-TS methods.

2. We evaluate FGAA on multiple tasks, showing that it outperforms existing activation steering methods in steering performance and steered output quality.

3. We investigate the impact of varying steering scales on the generalization capabilities of models across a diverse range of activation steering methods.

Our findings advance both theoretical understanding of LLM activation patterns and practical steering methodology.

## 2   RELATED WORK

**Mechanistic Interpretability and SAEs**   Bereska and Gavves (Bereska & Gavves, 2024) outlined the central hypothesis of mechanistic interpretability: models learn human-comprehensible algorithms and can be understood, despite having no incentive to make these algorithms legible to humans during loss minimization. A key challenge in this field was identified by Scherlis *et al.* (Scherlis et al., 2022), who found that individual neurons often encode multiple distinct features (polysemanticity), making direct analysis of neuron behavior difficult. This is caused by superposition, the phenomenon of models representing more features than they have dimensions (Elhage et al., 2022). Sparse Autoencoders (SAEs) emerged as a solution to this challenge, with Cunningham *et al.* (Huben et al., 2024) demonstrating that SAEs could extract interpretable features from these superposed representations in transformer models. Bricken *et al.* (Bricken et al., 2023) further showed how these extracted features could be manipulated during inference to affect model behavior. Our work uses SAEs to extract interpretable features from different inputs, to construct a set of desired SAE features to steer for.

**Linear Representation Hypothesis**   Park *et al.* (Park et al., 2023) introduced the Linear Representation Hypothesis, showing that neural networks encode high-level concepts linearly in their representation spaces. Several studies support this hypothesis: the extraction of linear features using SAEs (Bricken et al., 2023), the effectiveness of linear probes in detecting features in the residual stream (Chanin et al., 2024), and the results from activation steering methods. We leverage this linearity assumption in both our feature selection process and its use of linear effect approximators to optimize steering vectors.

**Activation Steering**   Turner *et al.* (Turner et al., 2024) introduced activation steering (or activation engineering) to influence LLM behavior by modifying model activations during inference. Building on this work, Panickssery (Rimsky et al., 2024) introduced CAA, which computes steering vectors by averaging the difference in residual stream activations between sets of positive and negative examples of a particular behavior. Chalnev *et al.* (Chalnev et al., 2024) developed linear effect approximators, a linear function that predicts how steering vectors affect SAE features, allowing for targeted steering vector construction with reduced side effects. In our work, we apply the effect approximator framework to optimize CAA-derived steering vectors which are represented as SAE features.

## 3   FEATURE GUIDED ACTIVATION ADDITIONS

FGAA enhances CAA by operating directly in the SAE's latent space and employing optimization techniques to create more effective and coherent steering vectors. Our method consists of several key components that work together to identify and utilize the most relevant activation patterns while minimizing unwanted effects. For the rest of this paper, in the interest of clarity, positive and negative examples of a particular behavior used in CAA are termed as desired and undesired examples, while features refer to SAE latents.

## 3.1 SAE-BASED CONTRASTIVE ANALYSIS

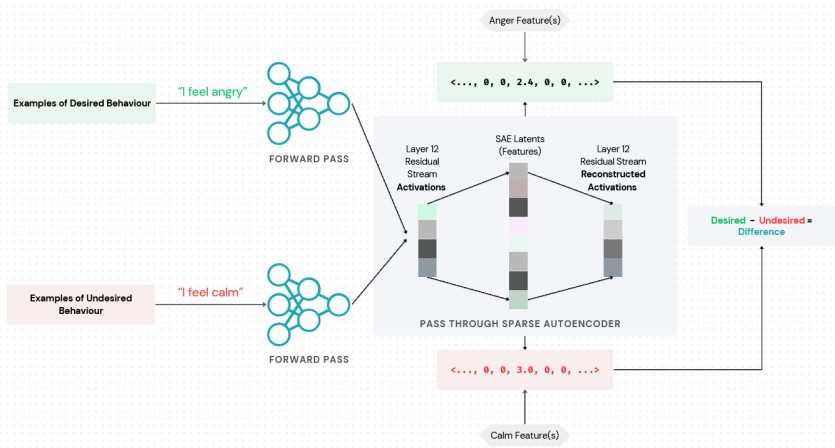

Figure 1: Diagram showing the process for computing $\mathbf{v}_{\text{diff}}$ on a simplified "Anger" task.

Unlike traditional CAA which operates on raw activations, FGAA computes contrastive differences in the SAE activation space. Given sets of positive and negative examples $X^+$ and $X^-$ which exhibit desired and undesired behaviors respectively, and an SAE with encoder $f$, we compute the difference vector as:

$$\mathbf{v}_{\text{diff}} = \frac{1}{|X^+|} \sum_{x \in X^+} f(h_l(x)) - \frac{1}{|X^-|} \sum_{x \in X^-} f(h_l(x)) \tag{1}$$

where $h_l(x)$ represents the hidden state activations at layer $l$ for input $x$, and $f(h_l(x))$ represents the mean SAE feature activations across all tokens. This produces a vector in the SAE's latent space that captures the key differences between desired and undesired behavior in terms of interpretable features.

## 3.2 FEATURE FILTERING

We apply three critical filtering steps to transform the difference vector into the target vector:

1. **Density Filtering**: We zero out features with activation density above a threshold $\theta$:

$$\mathbf{v}_{\text{filtered}}(i) = \begin{cases} 0 & \text{if } \rho(i) > \theta \\ \mathbf{v}_{\text{diff}}(i) & \text{otherwise} \end{cases} \tag{2}$$

   where $\rho(i)$ is the activation density of feature $i$ and $\theta = 0.01$ in our implementation.

2. **BOS Feature Removal**: We zero out features that activate most strongly on the Beginning Of Sequence (BOS) token:

$$\mathbf{v}_{\text{filtered}}(i) = \begin{cases} 0 & \text{if isBOS}(i) \\ \mathbf{v}_{\text{filtered}}(i) & \text{otherwise} \end{cases} \tag{3}$$

   where isBOS$(i)$ identifies features that have the highest activations at the BOS token. For Gemma family models, they are represented as `<bos>`.

3. **Top-k Selection**: Based on feature activation values, we retain the $n_1$ most positively activating and $n_2$ most negatively activating features:

$$\mathbf{v}_{\text{target}} = \text{concat}(\text{top}_{n_1}(\mathbf{v}_{\text{filtered}}), \text{top}_{n_2}(-\mathbf{v}_{\text{filtered}})), \quad n_1, n_2 \in \mathbb{Z}^+ \tag{4}$$

The three filtering steps in FGAA were developed through empirical observation of feature activation patterns across multiple steering tasks. Density filtering addresses a common issue where high-density features (those that activate frequently across many inputs) tend to dominate the difference vector despite their limited task specificity. By filtering out features with activation density above $\theta = 0.01$, we ensure the steering vector focuses on more specialized features that better characterize the target behavior. Similarly, BOS feature removal was implemented after observing a family of features that exclusively had the strongest activation on the BOS token (Appendix G), which often introduced artifacts in generation while contributing little to the desired steering effect. These features typically encode general linguistic patterns rather than task-specific behaviors. Finally, the selection of top $n_1$ positive and $n_2$ negative features helps eliminate noise from weakly activated features, focusing the steering vector on the most significant behavioral indicators.

### 3.3 LINEAR APPROXIMATOR OPTIMIZATION

We employ effect approximators (Chalnev et al., 2024) to solve for the optimal steering vector to produce the desired feature effects in $\mathbf{v}_{\text{target}}$. The linear effect approximator can be represnted as a function $\hat{\boldsymbol{y}} = \boldsymbol{x}M + \boldsymbol{b}$, where $\boldsymbol{x}$ is the $d_{\text{model}}$-dimensional steering vector, $M$ is a $d_{\text{model}} \times d_{\text{sae}}$ matrix, $\boldsymbol{b}$ has dimension $d_{\text{sae}}$, and $\hat{\boldsymbol{y}}$ is the predicted steering effects vector of dimension $d_{\text{sae}}$.

The approximator consists of a weight matrix $W$ and bias vector $\mathbf{b}$. Given our desired feature vector $\mathbf{v}_{\text{target}}$, we compute the optimized steering vector $\mathbf{v}_{\text{opt}}$:

$$\mathbf{v}_{\text{opt}} = \frac{W\mathbf{v}_{\text{target}}}{\|W\mathbf{v}_{\text{target}}\|} - \frac{W\mathbf{b}}{\|W\mathbf{b}\|} \tag{5}$$

For our implementation, $\mathbf{v}_{\text{target}}$ is L1 normalised for this calculation for consistent scaling of the relevant features, which helps maintain stable steering effects regardless of the magnitude of the original target vector.

### 3.4 FINAL STEERING APPLICATION

The final FGAA steering vector is applied to the model's hidden state at layer $l$ during generation:

$$h_l = h_l + \alpha\mathbf{v}_{\text{opt}} \tag{6}$$

where $\alpha$ is a scaling factor which we refer to as steering scale.

## 4 EVALUATIONS AND DISCUSSION

### 4.1 EFFECTIVENESS OF FGAA FOR STEERING

For our evaluations, FGAA is implemented using a pre-trained Gemma Scope (Lieberum et al., 2024) SAE with 16,384 features for the residual stream at layer 12 for Gemma-2-2B and Gemma-2-9B models. We selected these two models due to both computational constraints and the availability of open pre-trained SAE weights. Similarly, we apply steering to the residual stream at layer 12 and utilize pretrained effect approximators from (Chalnev et al., 2024) for both Gemma models. We focus on layer 12 in our evaluation, as collecting training data for effect approximators is time-intensive and must be done separately for each layer. Additionally, only layer 12 approximators for the models above have been made publicly available.

We evaluate FGAA against existing steering methods using the evaluation framework from (Chalnev et al., 2024), employing gpt-4o-mini to assess both behavioral alignment and coherence on a 1-10 scale, which we then rescale to the range [0,1]. Let $B$ represent the behavioral score which measures steering target achievement, and $C$ represent coherence which evaluates semantic correctness post-steering (exact criterion in Appendix C). We define the Behavioral-Coherence Score (BCS) as:

$$\text{BCS} = B \times C, \quad B, C \in [0, 1] \tag{7}$$

We generate FGAA steering vectors using optimal $n_1$ and $n_2$ values found from a hyperparameter sweep in Appendix A1. Each steering vector is applied to the model by adding the steering vector to the residual stream at every token position, sampling 100 steered text completions, each 33 tokens long beginning with the open-ended prompt "`<bos>I think`". For fair evaluation, all steering vectors are L2 normalised before applied. The following are implementation details for the other steering methods.

**Contrastive Activation Addition** (CAA), defined as the mean difference of model activations between a set of desired and undesired examples, averaged over token positions and examples.

**SAE feature steering**, using the decoder vector of a single relevant SAE feature.

**SAE targeted steering** (SAE-TS), setting the same relevant SAE feature used for SAE feature steering as the only active feature in $\mathbf{v}_{\text{target}}$.

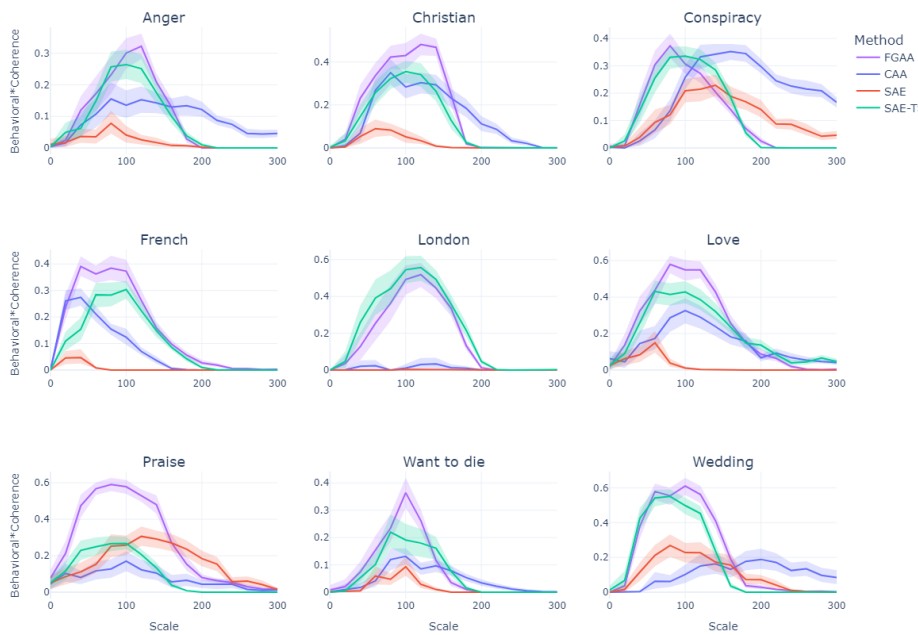

Figure 2: Plots showing mean BCS with 95% confidence intervals for the CAA, SAE, SAE-TS and FGAA steering methods on 9 tasks, for Gemma-2-2B.

Table 1: Mean BCS across steering methods on Gemma models. Best performing method per goal is underlined, best performing method on average in **bold**.

| | Gemma-2-2B | | | | Gemma-2-9B | | | |
|---|---|---|---|---|---|---|---|---|
| **Goal** | **CAA** | **SAE** | **SAE-TS** | **FGAA (Ours)** | **CAA** | **SAE** | **SAE-TS** | **FGAA (Ours)** |
| Anger | 0.1553 | 0.0778 | 0.2642 | 0.3220 | 0.2405 | 0.1622 | 0.2356 | 0.2116 |
| Christian | 0.3504 | 0.0896 | 0.3548 | 0.4815 | 0.3800 | 0.1736 | 0.3062 | 0.3640 |
| Conspiracy | 0.3523 | 0.2289 | 0.3356 | 0.3733 | 0.4195 | 0.2753 | 0.3202 | 0.4133 |
| French | 0.2743 | 0.0469 | 0.3035 | 0.3909 | 0.3235 | 0.3294 | 0.3909 | 0.4405 |
| London | 0.0331 | 0.0035 | 0.5570 | 0.5185 | 0.0519 | 0.1084 | 0.3407 | 0.3430 |
| Love | 0.3262 | 0.1494 | 0.4316 | 0.5798 | 0.3795 | 0.1072 | 0.2877 | 0.5437 |
| Praise | 0.1699 | 0.3062 | 0.2679 | 0.5914 | 0.2519 | 0.4247 | 0.5383 | 0.5785 |
| Want to die | 0.1311 | 0.0933 | 0.2198 | 0.3642 | 0.1449 | 0.1696 | 0.1294 | 0.1269 |
| Wedding | 0.1886 | 0.2681 | 0.5506 | 0.6101 | 0.2647 | 0.2896 | 0.5714 | 0.5595 |
| **Average** | 0.2201 | 0.1404 | 0.3650 | **0.4702** | 0.2729 | 0.2267 | 0.3467 | **0.3979** |

Table 1 demonstrates FGAA's superior performance across most tasks in the Gemma-2-2B model, while exhibiting heterogeneous effectiveness in the larger Gemma-2-9B architecture. FGAA achieves optimal performance in 8 out of 9 tasks for the 2B model, with notable improvements in semantic steering tasks such as 'Praise' and 'Love'. However, the performance distribution shifts substantially in the 9B architecture, where steering effectiveness is more evenly distributed among methods. Notably, CAA demonstrates superior performance in sentiment-based tasks. This pattern could suggest that FGAA's effectiveness exhibits non-linear scaling characteristics with model size.

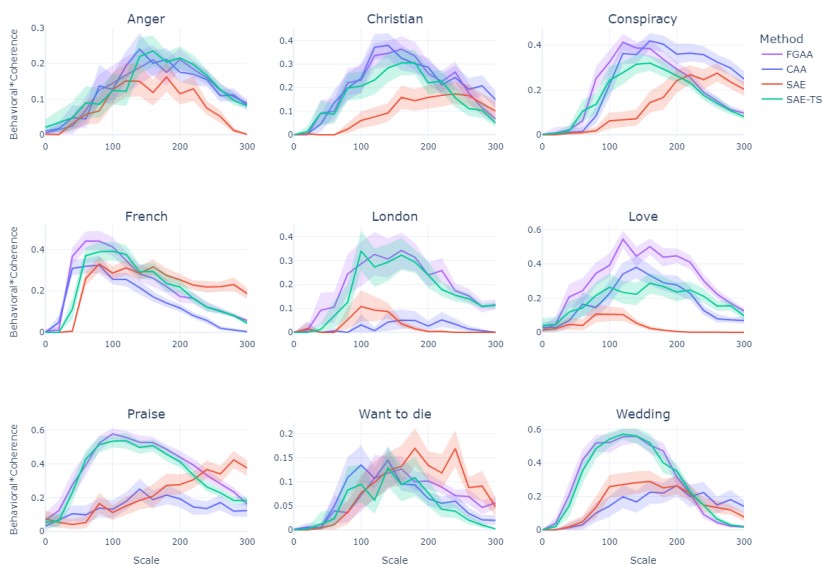

Figure 3: Plots showing mean BCS with 95% confidence intervals for the CAA, SAE, SAE-TS and FGAA steering methods on 9 tasks, for Gemma-2-9B.

**Advantages over Existing Methods**    FGAA addresses key limitations of current steering approaches:

- **Programmatic Feature Selection**: SAE-TS and SAE methods requires manual selection of a single feature to steer towards. FGAA programmatically identifies a spectrum of relevant features, while preserving the relationships in magnitude between them (refer to Table A.1 for an example). This is more realistic as, especially in lower width SAEs, it cannot be expected that every concept the LLM learns be cleanly encoded as an SAE latent. The presence of polysemantic and uninterpretable features extracted from SAEs across varying widths and models shows strong evidence for this, prompting research into Meta-SAEs (Anonymous, 2025) to further break down superposition. Instead, by representing concepts as a target vector in the feature space, we are able to achieve more precise concept representation. In larger width SAEs, this automated feature selection becomes more helpful due to the phenomena of feature splitting (Chanin et al., 2024), where a feature represented in a single latent in a smaller SAE can split into two or more latents in a larger SAE. FGAA systematically handles such cases by programmatically determine the relative steering magnitudes between semantically similar features. FGAA also handles the rare case where only targeting a single feature is the most effective steering approach, as detailed in Appendix D.

- **Interpretability**: While current CAA methods operate in opaque activation spaces, FGAA's backwards approach—determining desired effects in feature space before constructing steering vectors—provides explicit control over which features are steered, and to what extent. Through automatic interpertability (Paulo et al., 2024), SAE features can be labelled with human-interpretable descriptions (examples in Appendix B), allowing practitioners

to directly understand which semantic aspects of the model's behavior are being modified during steering. This transparency also allows us to filter away redundant components of the steering vector (via methods in Section 3.2) which would otherwise be present in CAA-derived vectors, allowing for more precise steering interventions.

## 4.2 EFFECTS OF STEERING ON GENERAL MODEL CAPABILITIES

We evaluate the impact of steering methods on model capabilities through perplexity testing on the OpenWebText (Gokaslan & Cohen, 2019) dataset and performance on MMLU (Massive Multitask Language Understanding) (Hendrycks et al., 2021) and MMLU-Pro (Wang et al., 2024) benchmarks. MMLU is a comprehensive evaluation benchmark that tests AI models using multiple choice questions spanning 57 different subjects, from STEM fields to humanities and social sciences. While the original MMLU primarily focuses on testing factual knowledge, MMLU-Pro builds upon this foundation by introducing more complex questions that require deeper reasoning abilities and increases the number of possible answers from 4 to 10 per question.

For perplexity evaluation, we use a sample of 100 records from OpenWebText, evaluating using steering vectors derived from the 9 steering tasks in Table 1. For MMLU and MMLU-Pro evaluations, we use fixed subsets of questions to ensure consistent comparison across steering methods: the first 5 questions from each subject category in MMLU, and the first 10 questions from each category in MMLU-Pro. Due to computational constraints, we limit these benchmark evaluations to steering vectors from 3 representative tasks in Table 1: Anger, Christian Evangelist, and Conspiracy. All experiments use Gemma-2-2B with steering vectors applied at layer 12 of the residual stream.

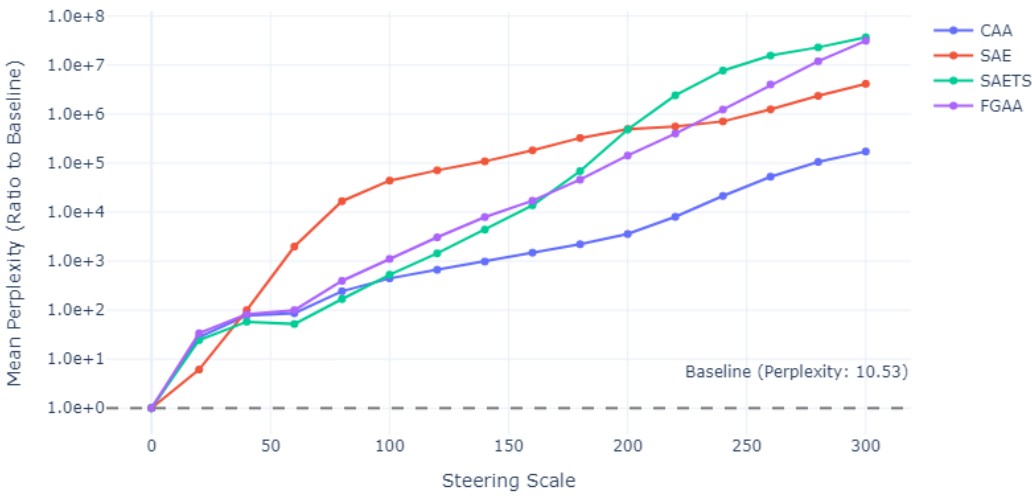

Figure 4: Relative perplexity vs steering scale (0-300). Lower values indicate better preserved language modeling. Results averaged across steering vectors from 9 different tasks, evaluated on the first 100 records in OpenWebText.

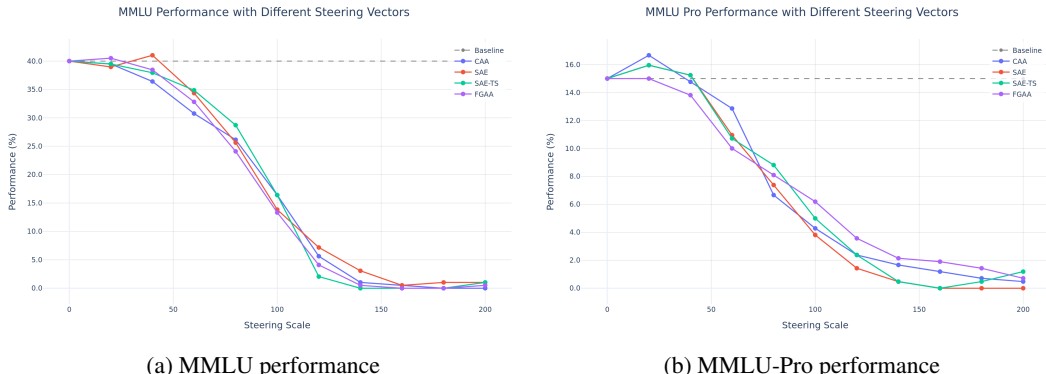

(a) MMLU performance            (b) MMLU-Pro performance

Figure 5: Benchmark performance vs steering scale (0-200). Higher values indicate better capability preservation. Results averaged across steering vectors from 3 tasks (Anger, Christian Evangelist and Conspiracy).

Figure 4 shows perplexity results across steering scales from 0 to 300, highlighting several critical insights. In the early-stage range (0-40), SAE's direct feature manipulation proves notably aggressive, while other methods maintain closer adherence to baseline performance. All methods demonstrate a distinct inflection point around scale 40, suggesting a universal threshold where steering begins to significantly impact model capabilities. We caution against drawing strong conclusions from high-scale (>150) behavior as all methods produce absurdly incoherent output in this range.

This degradation pattern is further corroborated by benchmark performance on MMLU and MMLU-Pro (Figures 5a, 5b). Both benchmarks demonstrate that model capabilities are largely preserved at lower steering scales but deteriorate as steering intensity increases. At scales below 50, all methods maintain close to baseline performance. However, beyond this threshold, we observe a consistent pattern of degradation across all steering approaches, with performance declining sharply between scales of 50 and 150 before converging near zero at higher scales.

These findings highlight an important trade-off in activation steering: while lower steering scales (<50) allow for behavioral modifications while preserving model capabilities, stronger steering interventions come at an increasing cost to general model performance. The similar degradation patterns show that this trade-off must be considered regardless of steering method.

An intriguing observation is the slight increase in MMLU-Pro performance at low steering scales for CAA, SAE-TS, and SAE methods. This phenomenon may be analogous to how low levels of noise can enhance LLM inference performance, similar to effects observed with techniques like NEFTune (Jain et al., 2024). At very low steering scales, these steering vectors might function as beneficial noise that temporarily improves model capabilities before the more disruptive effects of steering become dominant at higher scales. The absence of this initial performance bump in FGAA, which instead shows stable performance, suggests its steering interventions are more precisely targeted. This aligns with FGAA's design objective of creating focused steering interventions through feature space optimization rather than introducing broader activation perturbations. While this observation merits further investigation to fully understand the underlying mechanisms, such analysis falls outside the scope of this paper.

## 5 LIMITATIONS

Our current approach relies heavily on the quality of feature extraction by the underlying SAE, and performance could potentially improve with advances in SAE architectures that achieve more precise monosemantic feature separation. The method's effectiveness may be limited by the SAE's ability to capture complex and atomic concepts in its latent space, particularly for abstract or nuanced steering tasks.

The optimal selection of $n_1$ and $n_2$ parameters appears to be task-dependent, making it challenging to establish universal guidelines for parameter selection. Also, developing metrics to evaluate the

effectiveness of our feature filtering methods proves to be a challenging task due to the qualitative nature of interpreting features.

## 6 FUTURE WORK

Future work could proceed along several promising directions. First, investigating how SAE width and quality of SAE features affects steering performance with FGAA could help establish optimal feature space dimensionality for general steering tasks. In addition, exploring techniques to minimize capability degradation at higher steering scales while maintaining steering effectiveness would address one of the key challenges identified in our experiments.

We believe the most promising direction to pursue would be applying FGAA to existing works in the activation steering space, to see if FGAA performance improvements carry over to safety tasks such as controlling sycophancy, hallucination and refusal in RLHF models (Rimsky et al., 2024) and reducing their social biases (Durmus et al., 2024).

## 7 CONCLUSION

This work introduced FGAA, a novel approach that combines CAA with insights from SAE representations to improve steering effectiveness in language models. Our evaluations demonstrated that FGAA achieves superior performance compared to existing steering methods across multiple tasks, particularly for the Gemma-2-2B model where it outperformed baselines in 8 out of 9 steering tasks. The method's success highlights the value of operating directly in interpretable feature spaces while maintaining precision through systematic feature filtering and optimization.

Our analysis revealed important insights about activation steering in general: performance degrades notably above certain steering scales, and there exists a fundamental tradeoff between steering strength and preservation of model capabilities.

The development of FGAA represents a significant step forward in controlled text generation, offering both theoretical insights into activation patterns in LLMs and practical advances in steering methodology. While challenges remain in areas such as SAE quality optimization and parameter selection, the method's demonstrated effectiveness across multiple tasks and architectures provides a strong foundation for future research. Particularly promising directions include investigating SAE width effects, developing techniques to minimize capability degradation at higher scales, and exploring applications to safety-critical steering tasks. These advances in precise model control have significant implications for the development of more reliable and controllable language models, contributing to the broader goal of creating AI systems that can be effectively guided while maintaining their core capabilities.

## 8 ACKNOWLEDGEMENTS

We would like to express our sincere gratitude to our research mentors, Dr Tan Guoxian and Dr Yan Ming, for their guidance throughout our research. We are grateful to Mr Chan Kwang Wen for contributing OpenAI API credits that enabled our evaluations. Special thanks to Chen Guang, Mr Slava Chalnev and Mr Logan Riggs for their insightful discussions on SAEs and activation steering. We also thank Chen Guang for providing the necessary compute resources for our work.

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

APPENDIX

## A SELECTION OF $n_1$ AND $n_2$ IN TOP-K FILTERING

### A1 PERFORMANCE ANALYSIS

Our initial investigation examined both positive and negative feature selection for steering vectors. However, empirical analysis (Appendix A3) revealed that negative features often degraded performance and produced inconsistent results (at least for the 9 tasks we evaluate on). This finding led us to simplify our approach to focus exclusively on positive features, setting $n_2 = 0$ and optimizing only for $n_1$.

We conducted a hyperparameter sweep for optimal $n_1$ from values $[1, 8]$ for all nine steering tasks, as seen in Figures A.1 and A.2.

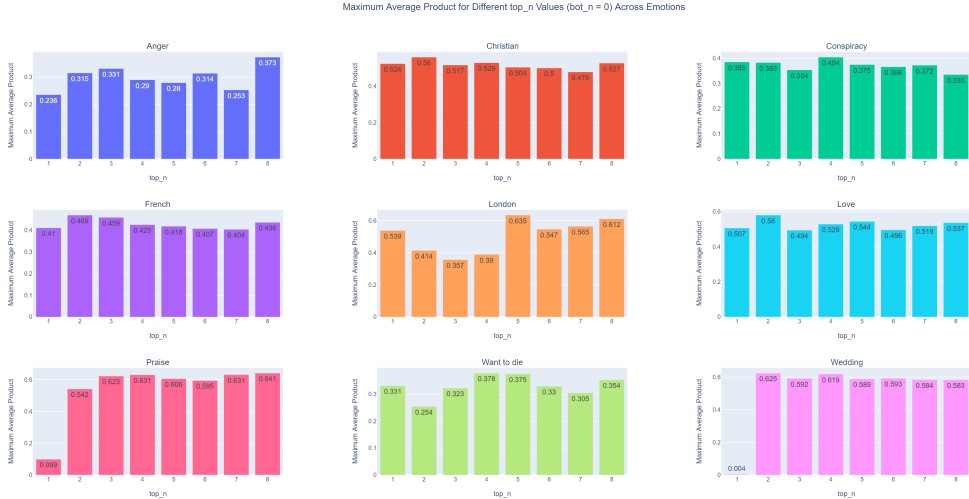

Figure A.1: Best mean BCS for different $n_1$ values ($n_2$=0) across 9 tasks, when steered on Gemma-2-2B. 30 samples generated for every $n_1$.

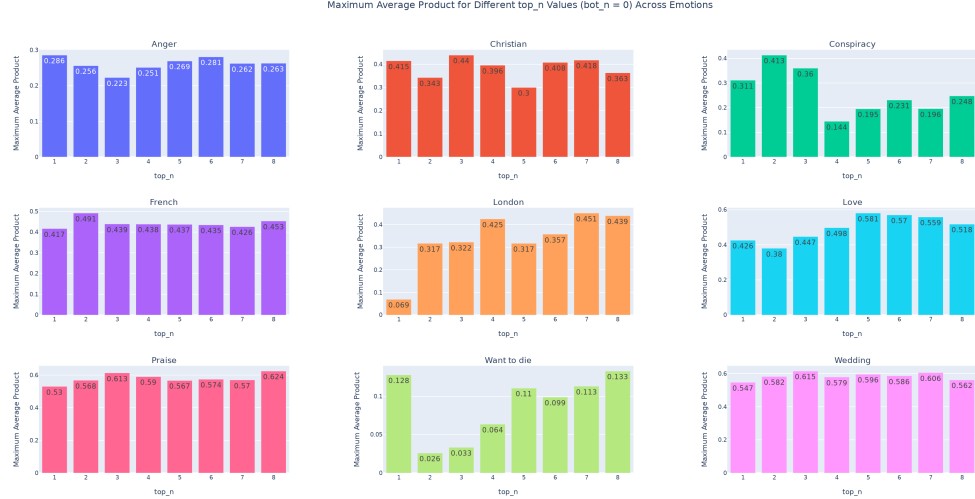

Figure A.2: Best mean BCS for different $n_1$ values ($n_2$=0) across 9 tasks, when steered on Gemma-2-9B. 30 samples generated for every $n_1$.

## A2 FEATURE ACTIVATION ANALYSIS

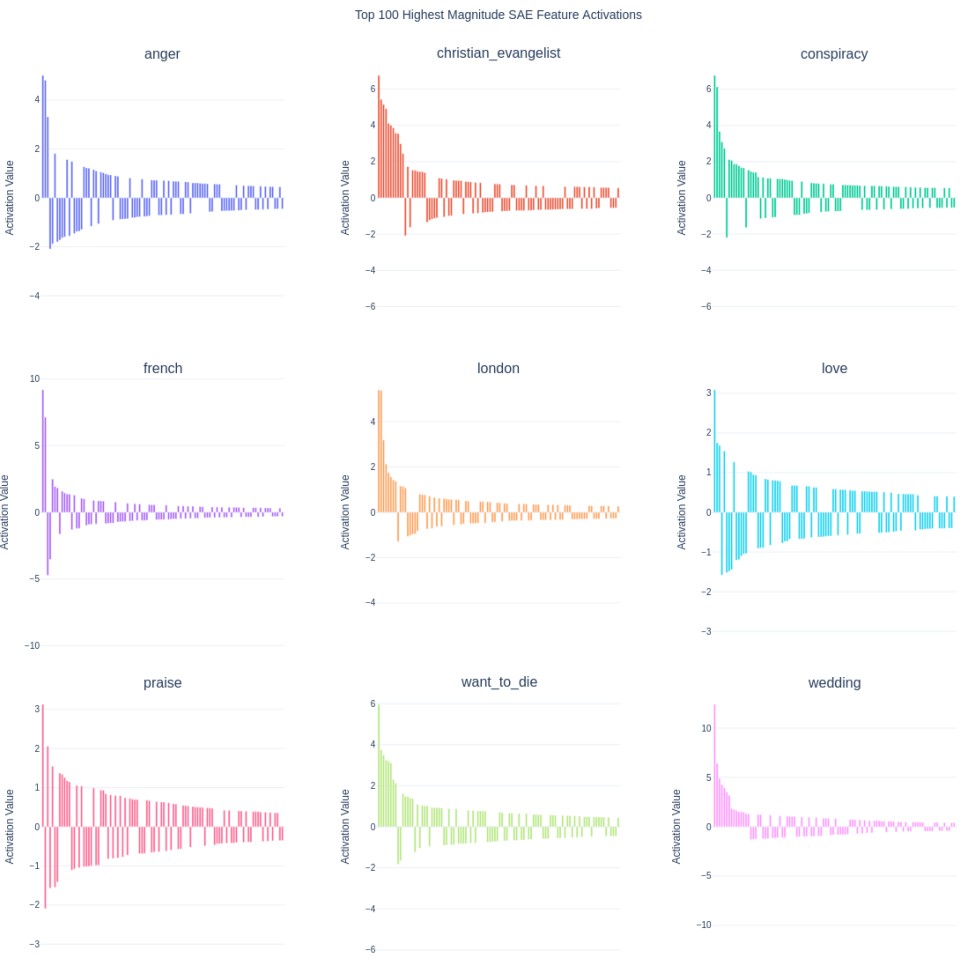

Figure A.3: Top 100 highest magnitude SAE feature activations across nine steering tasks, for Gemma-2-2B.

Referring to Figure A.3, the activation patterns show similarities in a few highly activating features, followed by many low activation features, which we hypothesise could indicate that the general semantic direction of the tasks can be captured succinctly with the few highest magnitude features.

This hypothesis is supported by performant steering in Table 1 with $n_1$ within the range $[1, 8]$, as well as Figure A.4 which shows diminishing gains in performance on Anger and Praise tasks when increasing $n_1$ past a certain point (E.g. for Praise task, this point seems to be in the range $[6, 11]$).

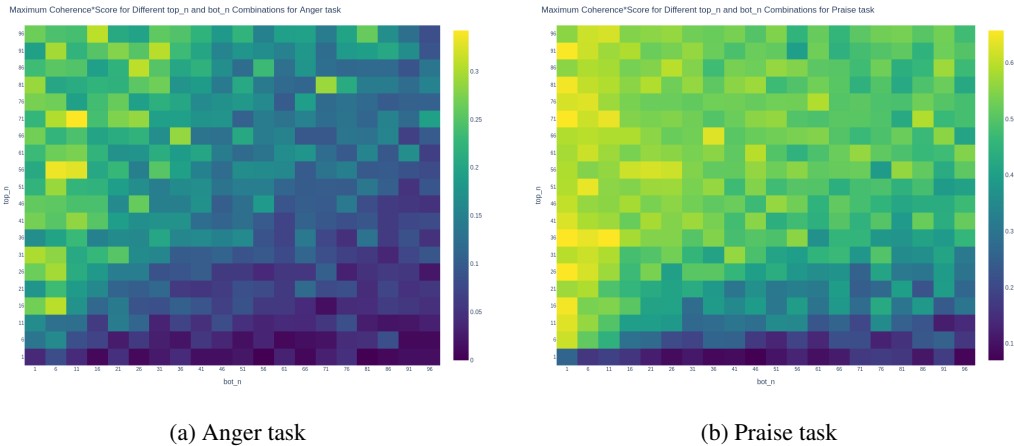

(a) Anger task                                  (b) Praise task

Figure A.4: Maximum Coherence*Score for different $n_1$ and $n_2$ combinations across Anger and Praise tasks, when steered on Gemma-2-2B. 10 samples generated for every combination of $n_1$ and $n_2$.

## A3  ANALYSIS OF NEGATIVE FEATURE EFFECTS

Table A.1: Features for "Praise" Target Vector for Gemma-2-2B ($n_1 = 10$, $n_2 = 10$)

| Positive Features | | |
|---|---|---|
| **Value** | **Index** | **Feature Description** |
| 3.130 | 4667 | Sentence starters and transitional phrases |
| 2.062 | 709 | Expressions of positive feedback and encouragement |
| 1.545 | 4267 | Positive adjectives and expressions of admiration |
| 1.373 | 3423 | Positive evaluations and recommendations |
| 1.338 | 1178 | Mathematical notation and statistical elements |
| 1.259 | 4248 | Phrases signifying quality and reliability |
| 1.177 | 12929 | Concepts of service and philanthropy |
| 1.148 | 10019 | Expressions of good wishes |
| 1.056 | 6668 | Exclamation marks and expressions of enthusiasm |
| 1.040 | 991 | Expressions of encouragement and validation |
| **Negative Features** | | |
| **Value** | **Index** | **Feature Description** |
| -2.093 | 13367 | Phrases conveying skepticism and criticism |
| -1.568 | 1024 | Phrases related to misbehavior |
| -1.545 | 9118 | Terms related to behavior changes |
| -1.415 | 4561 | Negative descriptors and crime terms |
| -1.108 | 11281 | Expressions of disappointment |
| -1.079 | 787 | Possessive pronouns |
| -1.047 | 15620 | Professional conduct elements |
| -1.021 | 15 | Expressions of humor and sarcasm |
| -1.019 | 718 | Expressions of emotional turmoil |
| -1.014 | 12851 | Expressions of fatigue and distress |

The observed performance degradation with increasing $n_2$ values at low $n_1$ reveals an important asymmetry in steering feature semantics. Analysis of feature distributions from Table A.1 shows that positive features typically form cohesive semantic clusters (e.g., encouragement, good wishes and positive feedback), while negative features exhibit broader semantic diversity (e.g., references to crime and expressions of humor). This asymmetry appears inherent to the nature of concept representation: while positive instances of a concept cluster around specific semantic elements, negative instances encompass a vastly larger semantic space of alternatives.

This semantic disparity explains why increasing $n_2$ diminishes steering effectiveness. Including too many negative features risks suppressing a broad range of linguistic patterns potentially necessary for coherent text generation. Additionally, consistently poor steering performance from Figure A.4 at low $n_1$ values and high $n_2$ values suggest that avoidance-based steering through negative features may be inherently less effective in LLMs than positive feature guidance.

We also find emperically that negative features are highly sensitive to the selection of examples with non-desired behavior.

## B    EXAMPLES OF CONSTRUCTED FILTERED TARGET VECTORS

Explanations for each feature taken from Neuronpedia. Each explanation is generated through automatic interpretation, by showing the top activations to `gpt-4o-mini` and asking it to explain what it thinks this feature is about.

### B1    CONSPIRACY GEMMA-2-9B

Desired examples:

```
"The government is hiding the truth about alien contact"
"A secret cabal of elites controls the world's economy"
"The media is spreading disinformation to manipulate
    public opinion"
"Whistleblowers are being silenced to cover up a
    massive scandal"
"A shadow government is pulling the strings behind the scenes"
"The New World Order is implementing their sinister plan"
"The elites are controlling everything"
"The deep state"
"Lizzard people control the government"
"Connect the dots."
```

Undesired examples:

```
"Reputable sources have fact-checked and debunked that claim"
"Transparency in government helps prevent corruption and abuse"
"Peer-reviewed studies provide reliable,
evidence-based conclusions"
"The weather forecast says it might rain tomorrow"
"My phone battery doesn't last as long as it used to"
"That new movie got good reviews"
"My friend got a job in government"
"He's at school today"
"I plan to stop by the store"
"Let's play secret santa"
```

Table B.1: Features for "Conspiracy" Target Vector for Gemma-2-9B ($n_1 = 5$, $n_2 = 5$)

| Positive Features | | |
|---|---|---|
| **Value** | **Index** | **Feature Description** |
| 5.378 | 3358 | References to government, intelligence agencies, and organized crime |
| 5.165 | 11032 | Terms related to political correctness and liberal ideologies |
| 4.122 | 569 | References to crime, corruption, and political manipulation |
| 3.941 | 1456 | Actions related to processing or interpreting information |
| 3.613 | 4668 | Occurrences of the word "the" |
| 3.414 | 2361 | Terms related to political and economic power struggles |
| 2.896 | 7379 | Mentions of political or legal actions related to public safety |
| **Negative Features** | | |
| **Value** | **Index** | **Feature Description** |
| -2.128 | 12407 | Terms related to legal or contractual language |
| -1.778 | 11912 | Questions and inquiries about information or assistance |
| -1.746 | 1188 | References to evidence-based practices and research |
| -1.714 | 6013 | Phrases that express a call to action or commands |
| -1.651 | 4358 | Expressions of personal experience and storytelling |
| -1.650 | 3685 | Descriptions of weather conditions and their effects |

Rollouts at Scale = 120 (Optimal Scale):

```
"<bos>I think this is a photo of a giant squid
    attacking a Russian submarine,
    and it is one of the most Incredible Aliens captured
    in Antarctica! These mind"
"<bos>I think the only reason that Donald Trump has
    been exposed is the Pentagon's secret Advanced Aerospace
    Program called the Advanced Space Agency (AASPA)
    and the US"
"<bos>I think there are many planets out There in
    our Universe. But, there may be Alien Lifeforms
    on Mars that could be Behind The Pyramids on Mars. "
"<bos>I think it is high Time for the disclosure
    of Extraterrestrial Technology to the World.
    Recently, a group of British researchers
    have claimed that they have uncovered"
"<bos>I think the FBI has captured
    a 'Super-Being' named Ra from an Alien Craft
    that fell to Earth 6,000 years Ago, according"
"<bos>I think that we are being manipulated by
    Extraterrestrial Races through The Vatican
    and the Catholic Church, so I began an ambitious
    Project to uncover the truth. "
```

## B2  LOVE (GEMMA-2-2B)

Desired examples:

```
"I feel an overwhelming sense of love and affection for you"
"Your kindness and compassion fill my heart with adoration"
"I'm devoted to you and cherish every moment we spend
    together"
"The warmth of your embrace makes me feel completely at home"
"My admiration for you grows stronger with each passing day"
"I'm passionate about our relationship and excited for our
    future"
"I love you"
"Love"
"You are the light of my life and my reason for smiling"
"You're my everything"
```

Undesired examples:

```
"I can't stand being around you, it fills me with resentment"
"Your actions have made me lose all respect for you"
"I feel nothing but disdain when I think about our past"
"The mere thought of you fills me with intense dislike"
"I've grown to despise everything about this situation"
"Your presence brings out feelings of animosity in me"
"I don't care"
"Hate"
"I feel absolutely nothing for you"
"You mean nothing to me"
```

Table B.2: Features for "Love" Target Vector for Gemma-2-2B ($n_1 = 10$, $n_2 = 10$)

| Positive Features | | |
|---|---|---|
| **Value** | **Index** | **Feature Description** |
| 3.090 | 7863 | Instances and expressions of love |
| 1.754 | 4990 | Expressions of love and emotional connections |
| 1.690 | 5679 | References to speaker's personal experiences |
| 1.657 | 10543 | Coordinating conjunctions connecting clauses |
| 1.546 | 2623 | References to personal accountability |
| 1.369 | 13074 | Phrases related to physical intimacy |
| 1.269 | 14739 | References to romantic relationships |
| 1.231 | 16036 | Expressions of love and enjoyment |
| 1.091 | 15596 | Forms of the verb "to be" in various tenses |
| 1.032 | 15995 | Possessive pronouns indicating ownership |
| **Negative Features** | | |
| **Value** | **Index** | **Feature Description** |
| -1.584 | 9781 | Expressions of indifference or lack of concern |
| -1.524 | 13367 | Phrases conveying skepticism or criticism |
| -1.487 | 3869 | Negative sentiments and expressions of disdain |
| -1.446 | 13803 | Phrases expressing negation or absence |
| -1.376 | 16253 | Phrases expressing skepticism or doubt |
| -1.206 | 9084 | Phrases related to systemic issues |
| -1.196 | 1369 | Terms related to horror and negative experiences |
| -1.103 | 870 | Expressions of discomfort or well-being |
| -1.055 | 2547 | Instances of "me" in different contexts |
| -1.039 | 2605 | References to presence or absence of evidence |

Rollouts at Scale = 80 (Optimal Scale):

```
"<bos>I think every girl and woman knows what it feels
    like to have a crush. At one point or another, we
    have all been loved so hard by a man"
"<bos>I think it is always difficult for parents to see their
    girls happy and in love! @laurenbush is the kindest, most
    beautiful soul who blessed our world"
"<bos>I think this has to be the most amazing
    and inspirational story. I love my parents,
    and there is no one more special to me. The best"
"<bos>I think it's so important to capture these
    little moments with my love. I think she is
    so beautiful. She is 1 week older than my sweet"
"<bos>I think it's safe to say that we simply love
    all of Justin and Hailey's wedding photos
    and it's no wonder why we always swoon"
"<bos>I think the best part about dating is being your
    own best friend. I wish I knew how much I loved you
    in a different way, and the way"
```

## C  STEERING EVALUATION CRITERION

### C1  SCORING PROMPT STRUCTURE

The evaluation process utilizes `gpt-4-mini` with the following standardized prompt structure:

> You score texts generated by a language model based on the following criterion: {{criterion}}
> You provide a score from 1 to 10.
> The language model was given a prompt and generated the following text. Evaluate the text based on the criterion. Output format should be JSON with the following fields: "score" (int)

### C2  COHERENCE CRITERION

All tasks are evaluated against the following coherence criterion:

> The text is coherent and the grammar is correct.

### C3  TASK-SPECIFIC BEHAVIORAL CRITERION

**Anger**

The text contains angry words or angry sentiment.

**Christian Evangelist**

The text contains discussion about christianity and its virtues

**Conspiracy**

The text contains language that suggests a conspiracy theory or belief.

**French**

The text contains french words and french language

**London**

Mentions London or anything related to London.

**Love**

The text contains words about love or contains the sentiment.

**Praise**

The text contains language that bestows praise.

**Want to Die**

The text contains language that suggests a person wants to die.

**Wedding**

The text contains discussion about a wedding.

## D  COSINE SIMILARITY OF STEERING VECTORS

Table D.1: Cosine similarity between FGAA vectors and other steering vectors across different methods and tasks. Higher values indicate greater similarity with FGAA direction.

| Gemma-2-2B | | | | Gemma-2-9B | | | |
|---|---|---|---|---|---|---|---|
| **Task** | **CAA** | **SAE** | **SAE-TS** | **Task** | **CAA** | **SAE** | **SAE-TS** |
| Anger | 0.1904 | 0.2056 | 0.9116 | Anger | 0.2052 | 0.4123 | 1.0000 |
| Christian | 0.2994 | 0.2410 | 0.9348 | Christian | 0.3365 | 0.0872 | 0.9628 |
| Conspiracy | 0.1824 | 0.2445 | 0.9259 | Conspiracy | 0.2267 | 0.2791 | 0.9487 |
| French | 0.4164 | 0.2813 | 0.9504 | French | 0.4093 | 0.2359 | 0.9219 |
| London | 0.2186 | 0.0523 | 0.9092 | London | 0.2264 | 0.1632 | 0.9528 |
| Love | 0.2678 | 0.1474 | 0.9394 | Love | 0.3293 | 0.1245 | 0.8976 |
| Praise | 0.1785 | 0.0578 | 0.7668 | Praise | 0.1989 | 0.1339 | 0.8842 |
| Want to die | 0.1712 | 0.2725 | 0.8283 | Want to die | 0.2038 | 0.1244 | 0.7970 |
| Wedding | 0.1309 | 0.2624 | 0.8610 | Wedding | 0.2438 | 0.3480 | 0.9904 |
| **Average** | 0.2284 | 0.1961 | 0.8919 | **Average** | 0.2644 | 0.2121 | 0.9284 |

Analysing Table D.1, SAE-TS vectors are nearly parallel to FGAA vectors (similarity >0.85) across almost all tasks in both models. This high alignment explains similar results between the two methods in Table 1, suggesting that FGAA and SAE-TS independently converge on similar steering solutions even though FGAA considers multiple features while SAE-TS targets just one. Identical steering vectors for the Anger task under Gemma-2-9B is due to selection of $n_1 = 1$ from our hyperparameter sweep, hence coincidentally only including the same feature selected for SAE-TS and SAE methods. In contrast, both CAA and single-feature SAE steering operate in substantially different directions, with similarities mostly below 0.3. This is particularly interesting for CAA, since FGAA builds upon its methodology — the low similarity suggests that FGAA's feature-space optimization via filtering and the effect approximator significantly alters the steering direction from raw activation differences.

# E   TRADE-OFF CURVES

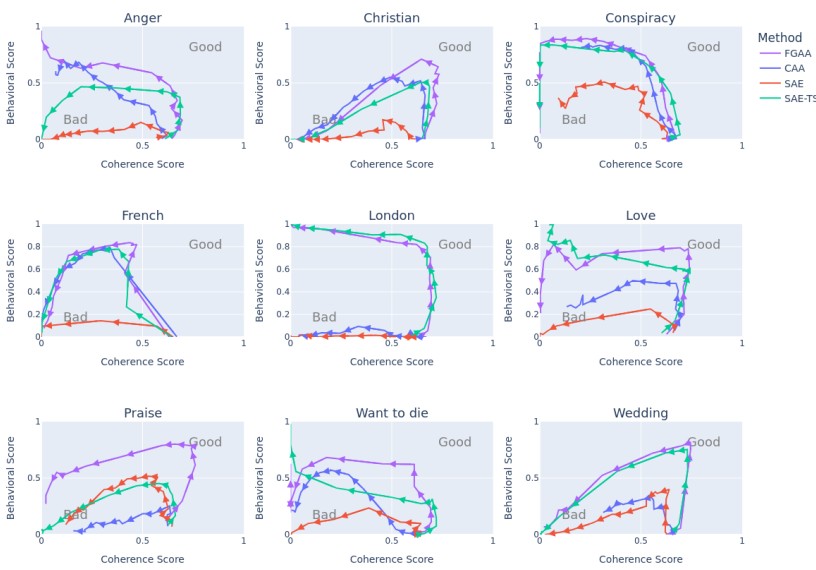

Figure E.1: Score trade-off curves for Gemma-2-2B, plotting both Coherence and Behavioral scores against increasing steering scale values. Each line tracks a distinct steering technique, with the optimal results appearing in the upper-right quadrant, where both Coherence and Behavioral metrics reach their highest values.

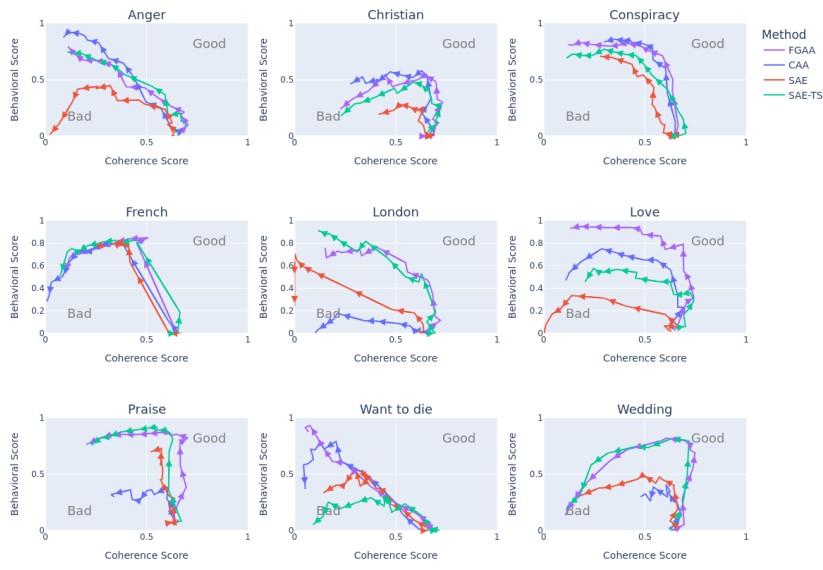

Figure E.2: Score trade-off curves for Gemma-2-9B, plotting both Coherence and Behavioral scores against increasing steering scale values. Each line tracks a distinct steering technique, with the optimal results appearing in the upper-right quadrant, where both Coherence and Behavioral metrics reach their highest values.

## F    Normalisation of $v_{target}$

As described in Section 3.3, we L1 normalise $\mathbf{v}_{target}$ prior to finding the optimal steering vector via the linear effect approximator function. Emperically, we find this produces better performing steering vectors than using L2 normalisation (when evaluated on the 9 tasks in Table 1), though we are unsure why. A possible theory is that L1 normalization's more equal treatment of features across different magnitudes helps preserve information from moderately activated features that might be overly suppressed by L2 normalization's quadratic scaling. Since L2 normalization is more sensitive to outliers and gives greater weight to larger values, it could potentially over-emphasize a few highly activated features while severely diminishing the contribution of moderately activated ones that still carry meaningful steering signal. L1 normalization's linear scaling might therefore better maintain the broader distribution of feature activations that emerges from our filtering process. This could also imply that the distribution of feature activations derived in $\mathbf{v}_{target}$ may not be entirely representative of the significance of the respective features in producing the steering goal. However, this observation remains empirical, and further investigation into understanding this phenomenon may provide a better understanding of SAE features for effective steering.

## G  FAMILY OF BOS FEATURES

Table G.1: Identified BOS Features from Gemma-2-2B 16k SAE (non-exhaustive). Descriptions marked with an asterisk (*) are the authors' interpretations. Uninterpretable features are not included.

| Index | Description |
|-------|-------------|
| 11087 | *the first token of a text |
| 3220 | *BOS token |
| 11752 | *BOS token |
| 12160 | *BOS and newline token |
| 11498 | *BOS token |
| 12110 | elements of numerical or mathematical notation |

