# OpenReview forum: "Interpretable Steering of Large Language Models with Feature Guided Activation Additions"
_ICLR.cc/2025/Workshop/BuildingTrust — BuildingTrust_

### Official Review · Reviewer_2gQ6 · 2025-02-27
**FGAA is a well motivated and promising improvement over SAE-TS and CAA**

**Rating:** 7
**Confidence:** 4

**Review:**

Summary: The paper introduces Feature Guided Activation Additions (FGAA), a new method for steering LLMs that builds upon CAA and SAE-TS.

Strengths:
1. FGAA is a novel and well motivated method. The three steps of feature filtering (density filtering, bos feature removeal, top-k selection) are explained and motivated.
2. FGAA has strong performance on both steering performance and output quality. The method narrowly beats SAE-TS, but clearly outperforms CAA and SAE.
3. FGAA is compared against alternative activation steering methods (CAA, SAE, SAE-TS). While other steering methods have been published recently, the comparison to the chosen steering methods is natural. The number of tasks (9), models (2) is sufficient for showing the potential of FGAA over its alternatives.
4. Using different Evaluation metrics for measuring steering method performance: BCS with gpt-4o-mini to assess both behavioral alignment and coherence. And also evaluate effects of steering on general model capabilities via measuring perplexity on OpenWebText, and performance on MMLU and MMLU-Pro.

Weaknesses:
1. For the paper to be considered stronger, more thorough comparisons of FGAA to CAA, SAE, SAE-TS and other alternatives need to be made. More tasks, models could be evaluated. Also tests for statistical significance of FGAA's superior performance over SAE-TS. The current comparison is sufficient for a workshop-level paper, but not for a full conference submission.
2. Visual presentation. The paper contains a lot of white space. Figure 1 should have larger fonts to be more readable. Table 1 goes over the linewidth. Figure 4 has a simple message but takes up a lot of space. Figure 5 should have a larger font size for legend and axis labels.
3. FGAA builds upon the main ideas from CAA and SAE-TS and is more of an incremental improvement by combining existing ideas. The results of FGAA are only marginally better for many tasks (Anger, London, Wedding, ...)

Recommendation: Good paper, accept. Overall, the paper introduces a novel steering method that is potentially an improvement over CAA and SAE-TS. More Analysis would be needed to clearly show its advantage over CAA and SAE-TS to make the paper more significant.

---

### Official Review · Reviewer_rayq · 2025-03-01
**Nice new steering method!**

**Rating:** 8
**Confidence:** 4

**Review:**

Proposes a novel activation steering method that is a pareto improvement on existing method by incorporating a lot of additional mechanisms that bridge gaps in past steering techniques.

**Quality**

Pros:
- FGAA shows superior performance in steering effectiveness and coherence degradation compared to past methods, usage of behavior-coherence score show consideration of real world usage,
- Was evaluated across the Gemma family and with a range of behaviors, further supporting effectiveness
- Systematic feature filtering and optimization

Cons:
- Rely heavily on quality of SAE features, which is currently a question in itself
- Optimal hyperparameters (like n1 and n2) look task dependent and can't extrapolate pattern.  Though this is true for all other steering techniques.
- Inconclusive results with negative features...?  Often degraded performance and produced inconsistent results

**Clarity**

- Method clearly outlined for every step!  Well supported by diagrams and formulations.

**Originality**

Pros:
- Novel method that leverages past results and techniques.  Targeting the feature space provides more fine grained control over steering target.
- Programmatically identified a spectrum of relevant features, unsees in previous steering techniques

Cons:
- An incremental advancement, not qualitatively different from other steering methods

**Significance**

Pros:
- A step forward in controller text generation for LLMs, maybe when we learn to extract better features, this can be applied more effectively in real life scenarios
- Programmatic feature selection is a major contribution to steering.

Cons:
- Very recent (<1 week) work suggest some degree of entanglement between broadly related features (ie 'good' things get grouped together same goes for the 'bad' things), it's the emergent misalignment paper.  If stronger models tend to have such structures, the promise of steering is put in question: "why do we need fine grained control over individual features if we can just tell the model to be good?".  Not inherently a flaw of this work but wanted to point out.

---

### Official Review · Reviewer_x7Ma · 2025-03-01
**Review of "Interpretable Steering of Large Language Models with Feature Guided Activation Additions"**

**Rating:** 4
**Confidence:** 4

**Review:**

**Summary:**

The paper introduces Feature Guided Activation Additions (FGAA), a proposed method for steering large language models (LLMs) by modifying their internal activations. FGAA aims to combine and improve upon two existing approaches: Contrastive Activation Addition (CAA) (Panickssery et al., 2023) and SAE-Targeted Steering (SAE-TS) (Chalnev et al., 2024).  CAA calculates steering vectors by contrasting activations from prompts exhibiting a desired behavior with those exhibiting an undesired (or opposite) behavior. SAE-TS, on the other hand, learns a linear mapping between steering vectors and their effects on the model's output, measured as changes in SAE feature activations. SAE-TS then use (inverse of) the mapping to get the steering vector to achieve the desired effect. FGAA attempts to integrate these two: it starts with the CAA method to initially form a effect vector (termed steering vector in the original of CAA framework), then get the steering vector using the SAE-TS approach (by applying the inverse of the mapping).  The authors also propose feature filtering steps (density filtering and BOS feature removal) and a hyperparameter search ($n_1$, $n_2$) to further refine the steering vector.

**Strengths:**

- **Algorithmic Contributions:** The specific implementation of the steering vector construction, including the filtering steps and the use of the effect approximators represent a novel contribution.
- **Empirical Evaluation:** The paper conducts evaluations using the same framework as Chalnev et al. (2024), comparing FGAA to CAA, SAE feature steering, and SAE-TS on several steering tasks. The results suggest that FGAA can outperform these baselines in some cases.
- **Investigates General Capabilities:** The authors analyze the impact of steering on model perplexity and performance on MMLU and MMLU-Pro benchmarks, which is important for understanding the trade-offs between steering and general language model capabilities.
- **Ablation and Hyperparameter Sweep:** The paper explores the influence of the $n_1$ and $n_2$ hyperparameters and provides other ablation. These discussions are diligent and insightful.

**Weaknesses:**

- **Ethical Concerns: Textual Similarity:** The paper exhibits substantial textual similarity to Chalnev et al. (2024), particularly in the introduction. While Chalnev et al. is cited, **the flow is almost identical** and it seems like text recycling, raising concerns about proper attribution and potential plagiarism. Several examples demonstrate this problem:
    1. **Opening Sentences / Motivation:**
        - **Chalnev et al.:** "There are widespread calls for better control of the behaviour of Large Language Models (LLMs; e.g. The White House (2023)). Current methods such as prompting (Wallace et al., 2024) and finetuning (Ouyang et al., 2022, Chung et al., 2022) offer some degree of control, but have clear limitations."
        - **This paper:** "Concerns are growing about effective, and reliable control of the behaviour of Large Language Models (LLMs), and they are being increasingly recognized. Conventional methods such as prompting (Wallace et al., 2024) and fine-tuning (Ouyang et al., 2022) provide a bit of control, yet they unfortunately have many important limitations that users must consider."
    2. **Limitations of Prompting/Fine-tuning:**
        - **Chalnev et al.:** "For example, prompting can be fragile and is often susceptible to methods that can subvert these instructions (Wei et al., 2023). Finetuning a model can be more robust but requires a curated dataset for training which can be both expensive and time-consuming to produce (e.g. Dubey et al. (2024))."
        - **This paper:** "Prompting is often weak and open to manipulation, but fine-tuning needs a lot of computing power and well-organized data."
    3. **Introducing Steering Vectors:**
        - **Chalnev et al.:** "*Steering vectors* (Turner et al., 2024) have the potential to be more robust than prompting, and both cheaper and easier to implement than finetuning. Steering vectors work by adding activations to the hidden state of a model, part way through the forward pass (Section 3)."
        - **This paper:** "A promising alternative is offered by activation steering, providing a stronger and more efficient approach than prompting and fine-tuning. This method adds steering vectors to the model’s hidden states and this influences its behavior during the forward pass."
    4. **Problem of Unpredictability:**
        - **Chalnev et al.:** "However, a problem with current steering methods is their unpredictability – it’s often unclear exactly how a steering vector will affect model behavior. Steering vectors may not produce the intended changes in the model’s output or may cause unforeseen behaviors, as we discuss in Section 3."
        - **This paper:** "...existing activation steering methods often miss one of precision, reliability and interpretability, leading to unintended model changes and poor output quality."

- **Misleading Branding as a Contrastive Method:** The paper frames FGAA as building on *contrastive* activation addition (CAA). However, the hyperparameter sweep reveals that the authors consistently set `n_2 = 0`. This means they *completely discard* the "negative" or contrastive component of the steering vector, effectively eliminating the contrastive aspect. The method, as implemented and evaluated, is *not* contrastive. This is a misrepresentation of the method.
- **Potential Lack of Novelty and Generalizability:** Because the method is not truly contrastive, its primary distinction from SAE-TS lies in averaging the SAE features of positive examples, rather than directly using the SAE feature's decoder vector (a single SAE feature). One may hypothesize that this is beneficial due to "feature splitting" (Chanin *et al.*, 2024), where a single concept might be represented by multiple SAE features. However, this raises a question:
    - **SAE-Specificity:** Is this benefit unique to SAEs exhibiting significant feature splitting? If a  SAE with less splitting were used, would FGAA still offer an advantage over SAE-TS? The paper does not address this, limiting the generalizability of the findings.
- **Task-Specific Prompt Engineering and Hyperparameter Tuning:** The evaluation relies on manually crafted prompts and task-specific hyperparameter tuning (finding optimal `n_1` values). This significantly limits the practicality and generalizability of the method. It's unclear how FGAA would perform on new tasks without extensive manual effort. The paper lacks a general, task-agnostic approach.

**Additional Note**

I would consider raising the rating if the ethical concern is deemed unfounded.

**References**

Chanin, D., Wilken-Smith, J., Dulka, T., Bhatnagar, H., & Bloom, J. (2024). *A is for Absorption: Studying Feature Splitting and Absorption in Sparse Autoencoders*. arXiv preprint arXiv:2409.14507.

Chalnev, S., Siu, M., & Conmy, A. (2024). *Improving Steering Vectors by Targeting Sparse Autoencoder Features*. arXiv preprint arXiv:2411.02193.

Panickssery, N., Gabrieli, N., Schulz, J., Tong, M., Hubinger, E., & Turner, A. M. (2023). *Steering Llama 2 via Contrastive Activation Addition*. arXiv preprint arXiv:2312.06681.

---

### Decision · Program_Chairs · 2025-03-04

Accept